REGISTERED REPORT

# Registered report: Discovery and preclinical validation of drug indications using compendia of public gene expression data

Irawati Kandela[1], Ioannis Zervantonakis[2], Reproducibility Project: Cancer Biology*

[1]Developmental Therapeutics Core, Northwestern University, Evanston, Illinois, United States; [2]Harvard Medical School, Boston, Massachusetts, United States

**Abstract** The Reproducibility Project: Cancer Biology seeks to address growing concerns about reproducibility in scientific research by conducting replications of 50 papers in the field of cancer biology published between 2010 and 2012. This Registered report describes the proposed replication plan of key experiments from 'Discovery and Preclinical Validation of Drug Indications Using Compendia of Public Gene Expression Data' by Sirota et al., published in *Science Translational Medicine* in 2011 (*Sirota et al., 2011*). The key experiments being replicated include Figure 4C and D and Supplemental Figure 1. In these figures, Sirota and colleagues. tested a proof of concept experiment validating their prediction that cimetidine, a histamine-2 (H2) receptor agonist commonly used to treat peptic ulcers (*Kubecova et al., 2011*), would be effective against lung adenocarcinoma (Figure 4C and D). As a control they also tested the effects of cimetidine against renal carcinoma, for which it was not predicted to be efficacious (Supplemental Figure 1). The Reproducibility Project: Cancer Biology is a collaboration between the Center for Open Science and Science Exchange, and the results of the replications will be published by *eLife*.

*For correspondence: fraser@scienceexchange.com

Group author details
Reproducibility Project: Cancer Biology
See page 7

## Introduction

In this paper, Sirota and colleagues tested their hypothesis that extant drugs could be repurposed to target alternative diseases; if so, this could improve efficiency in the search for new treatments. They compared data from the Gene Expression Omnibus (GEO)—which they used to determine gene expression signatures of diseases—to data from the Connectivity Map, which tracks the changes in mRNA expression caused by 164 drugs. By comparing these two mRNA expression sets, Sirota and colleagues created a similarity score to describe how similar the changes in mRNA expression were between each drug and each disease. They theorized that a similarity score close to −1 (exactly opposite signatures) might indicate that the drug could treat the disease.

In Figure 4C and D, Sirota and colleagues directly test their hypothesis by examining the effects of cimetidine, an H2 receptor blocker commonly used to treat gastric ulcers by reducing the production of stomach acid (*Kubecova et al., 2011*), on xenograft transplanted A549 lung adenocarcinoma cells. Mice treated with cimetidine showed a dose-dependent reduction in tumor size after 12 days of treatment. In Supplemental Figure 1, they also treated ACHN renal carcinoma cells with cimetidine, although cimetidine was not predicted to treat this cancer line. They observed no effect of cimetidine on the growth of this cancer cell line. These experiments will be replicated in Protocol 1. However, the conclusions that can be drawn from these experiments are limited by the fact that only a single cell line was tested with only a single drug.

To date, no direct replication of the experiments presented in Sirota and colleagues' Figure 4C and D or Supplemental Figure 1 has been reported. However, Stoyanov and colleagues did examine the effect of cimetidine on growth of A459 cells activated with histamine and reported that cimetidine did reduce proliferation in vitro (*Stoyanov et al., 2012*). An exploratory analysis of a cohort of diabetic patients demonstrated a decreased risk of developing lung cancer, specifically adenocarcinoma, in patients who took over-the-counter H2 receptor blockers, including cimetidine (*Hsu et al., 2013*).

## Materials and methods

Unless otherwise noted, all protocol information was derived from the original paper, references from the original paper, or information obtained directly from the authors. An asterisk (*) indicates data or information provided by the Reproducibility Project: Cancer Biology core team. A hashtag (#) indicates information provided by the replicating lab.

### Protocol 1: assessing the effect of cimetidine treatment on tumor growth in a xenograft model of lung carcinoma and a xenograft model of renal carcinoma

This protocol describes how to create xenograft tumors in severe combined immunodeficient (SCID) mice from A549 lung carcinoma cells (as seen in Figure 4C and D) or ACHN renal carcinoma cells (Supplemental Figure 1). Tumor growth is then assessed during 11 days of cimetidine treatment. Sirota and colleagues designed this experiment to test their predictions that A549 cells would be susceptible to cimetidine treatment while ACHN cells would not. Treatment with phosphate-buffered saline (PBS) alone will serve as the negative control, while treatment with the lung adenocarcinoma standard drug doxorubicin will serve as the positive control.

#### Sampling

- This experiment will use at least 12 mice per group for a final power of 82.4%.
  1. See 'Power calculations' for details.
- The experiment contains five cohorts total:
  1. A549 lung adenocarcinoma xenografts:
  a. Cohort 1: mice treated with PBS (negative control).
  i. N = 14.
  ■ To ensure at least 12 tumors develop.
  b. Cohort 2: mice treated with 2 mg/kg doxorubicin (Dox) (positive control).
  i. N = 5.
  ■ To ensure at least 3 tumors develop.
  c. Cohort 3: mice treated with 100 mg/kg cimetidine.
  i. N = 14.
  ■ To ensure at least 12 tumors develop.
  2. ACHN renal carcinoma xenografts:
  a. Cohort 1: mice treated with PBS (negative control).
  i. N = 14.
  ■ To ensure at least 12 tumors develop.
  b. Cohort 2: mice treated with 100 mg/kg cimetidine.
  i. N = 14.
  ■ To ensure at least 12 tumors develop.

#### Materials and reagents

| Reagent | Type | Manufacturer | Catalog # | Comments |
|---------|------|--------------|-----------|----------|
| PBS | Reagent | Invitrogen | 10010023 | |
| Fetal bovine serum | Reagent | Invitrogen | 16000-044 | |
| A549 cells | Cells | ATCC | #CCL-185 | Original unspecified |
| ACHN cells | Cells | ATCC | #CRL-1611 | Original unspecified |

*Continued on next page*

*Continued*

| Reagent | Type | Manufacturer | Catalog # | Comments |
|---|---|---|---|---|
| Hydrochloric acid (HCl) | Chemical | Sigma–Aldrich | 320331 | |
| Sodium hydroxide (NaOH) | Chemical | Sigma–Aldrich | 221465 | |
| Cimetidine | Drug | Sigma–Aldrich | C4522 | |
| Doxorubicin | Drug | Sigma–Aldrich | D1515 | |
| 4–6-week-old female SCID mice | Mice | Charles River | Strain code 236 | |
| EMEM | Media | Sigma | M2279 | |
| F-12 Ham's | Media | Sigma | N3520 | |
| Sodium pyruvate | Reagent | Sigma | S8636 | |
| Lipoic acid | Reagent | Sigma | T1395 | |
| Glutamine | Reagent | Sigma | 59202 | |

## Procedure

Notes:

- A549 cells are maintained in F-12 Ham's medium supplemented with 10% fetal bovine serum (FBS), 2 mM sodium pyruvate and 1 µM lipoic acid, based on ATCC recommendations.
  1. Lipoic acid is maintained as a 50 mg/ml stock in ethanol.
- ACHN cells are maintained in EMEM supplemented with 10% FBS, 2 mM glutamine and 1 mM sodium pyruvate, based on ATCC recommendations.
  1. All cells are grown at 37°C/5% $CO_2$.
- All cell lines will be sent for STR profiling and mycoplasma testing.

1. Culture A549 cells and ACHN cells.
2. Resuspend $5 \times 10^6$ cells in 100# µl PBS per injection.
3. Inject $5 \times 10^6$ cells (i.e., 100 µl of cell suspension) into the upper flank of 4–6-week-old* female SCID mice.
   a. Mice will be randomly assigned to receive injections with A549 cells or ACHN cells.
   i. Injections will be balanced so the total number of mice receiving A549 injections will be 33 and ACHN will be 28.
4. Measure tumor volume with calipers daily.
   a. Record daily tumor volume.
   b. Volume is defined as $mm^3 = 0.52 \times$ [width (cm)]$^2 \times$ height (cm).
   i. Mice shall be euthanized if they appear in undue distress according to the replicating lab's guidelines; if the animal has lost >20% body weight.
5. When tumor reaches a minimum of 100 $mm^3$ in volume (estimated time 2–3 weeks#), initiate treatment. Continue treatment for 11 days past this point.
   As each mouse reaches the injection criteria (i.e., 100 $mm^3$ tumor volume), randomly assign to a treatment group using the adaptive randomization approach with the time from injection of cells to when tumors reach at least 100 $mm^3$ and tumor volume at time of assignment as the covariates that are assessed as mice are sequentially assigned to a particular treatment group.
   i. Assignment will also take into account the pre-determined size of each treatment group.
   b. Treat mice by intraperitoneal injection according to cohort:
   i. A459 lung adenocarcinoma xenografts:
   1. Cohort 1: PBS (daily).
   2. Cohort 2: 2 mg/kg Doxorubicin (biweekly).
   3. Cohort 3: 100 mg/kg cimetidine (daily).
   ii. ACHN renal carcinoma xenograft injections:
   1. Cohort 1: PBS (daily).
   2. Cohort 2: 100 mg/kg cimetidine (daily).
   c. Continue daily tumor volume measurements.
6. Euthanize mice.
   a. Euthanize mice by $CO_2$ inhalation followed by cervical dislocation.
7. Harvest tumors and record weight (additional parameter).
   a. Image tumors alongside a ruler.

## Deliverables

- Data to be collected:
  1. Mouse health records, including age and tumor volume at start of injections, time of tumor detection, any excluded mice (including reason for exclusion).
  2. Raw data of tumor dimensions by day.
  3. Final weight of tumors.
  4. Graph of relative mean tumor weight in each cohort starting on Day 1 post-100 mm$^3$ (as seen in Figure 4C and Supplemental Figure 1).
     a. Normalize Day 2 onwards to the weight at Day 1.
  5. Image of all tumors alongside ruler (as seen in Figure 4D) for both A459 xenografts and ACHN xenografts.

## Confirmatory analysis plan

- Statistical analysis of replication data:
  1. At the time of analysis, we will perform the Shapiro–Wilk test and generate a quantile–quantile (q–q) plot to attempt to assess the normality of the data and also perform Levene's test to assess homoscedasiticity. If the data appear skewed, we will attempt a transformation in order to proceed with the proposed statistical analysis listed below and possibly perform the appropriate non-parametric test.
     a. Comparison of the mean relative tumor weight of 100 mg/kg cimetidine treatment at day 11 as compared to PBS treatment at day 11 for both A549 and ACHN xenograft tumors.
     i. Two-way analysis of variance (ANOVA) (2 × 2 factorial) followed by Bonferroni corrected Welch's $t$-tests for the following comparisons:
        - PBS-treated A549 tumors vs cimetidine-treated A459 tumors.
        - PBS-treated ACHN tumors vs cimetidine-treated ACHN tumors.
     ii. Additional comparison of PBS-treated A459 tumors to doxorubicin-treated tumors.
        - Bonferroni corrected Welch's $t$-test outside the framework of the ANOVA.
- Meta-analysis of original and replication attempt effect sizes:
  1. This replication attempt will perform the statistical analysis listed above, compute the effects sizes, compare them against the reported effect size in the original paper and use a meta-analytic approach to combine the original and replication effects, which will be presented as a forest plot.

## Known differences from the original study

- The replication attempt will encompass the PBS control, the doxorubicin control and the highest dose of cimetidine (100 mg/kg). It will not include the 25 mg/ml or 50 mg/ml cimetidine treatment groups.
- While the original study performed injections of 5 × 10$^6$ cells per microliter of PBS, on the advice of the replicating lab we will inject the same number of cells but suspended in 100 μl PBS.

## Provisions for quality control

Mice will be randomly assigned to xenograft model and treatment type. All data obtained from the experiment—raw data, data analysis, control data and quality control data—will be made publicly available, either in the published manuscript or as an open access dataset available on the Open Science Framework (https://osf.io/hxrmm/).

## Power calculations

For details on power calculations, please see analysis files on the Open Science Framework:

- https://osf.io/uazfe/wiki/home/

## Protocol 1

Note: data values estimated from published figures. Error bars assumed to represent SEM.

## Summary of original data

| Figure 4C: A549 xenograft tumor size | | Normalized mean weight | SEM | SD | N |
|---|---|---|---|---|---|
| PBS | Day 1 | 1 | 0.25 | 0.61 | 6 |
| | Day 2 | 1.28 | 0.25 | 0.61 | 6 |
| | Day 3 | 0.98 | 0.34 | 0.83 | 6 |
| | Day 4 | 1.35 | 0.24 | 0.59 | 6 |
| | Day 5 | 1.28 | 0.26 | 0.64 | 6 |
| | Day 6 | 1.39 | 0.24 | 0.59 | 6 |
| | Day 7 | 1.63 | 0.25 | 0.61 | 6 |
| | Day 8 | 1.98 | 0.24 | 0.59 | 6 |
| | Day 9 | 2.98 | 0.25 | 0.61 | 6 |
| | Day 10 | 2.83 | 0.31 | 0.76 | 6 |
| | Day 11 | 3.3 | 0.23 | 0.56 | 6 |
| Dox | Day 1 | 1 | 0.25 | 0.61 | 6 |
| | Day 2 | 0.96 | 0.12 | 0.29 | 6 |
| | Day 3 | 0.9 | 0.17 | 0.42 | 6 |
| | Day 4 | 0.87 | 0.12 | 0.29 | 6 |
| | Day 5 | 0.8 | 0.13 | 0.32 | 6 |
| | Day 6 | 0.94 | 0.14 | 0.34 | 6 |
| | Day 7 | 1.2 | 0.14 | 0.34 | 6 |
| | Day 8 | 1.63 | 0.21 | 0.51 | 6 |
| | Day 9 | 1.55 | 0.13 | 0.32 | 6 |
| | Day 10 | 1.84 | 0.14 | 0.34 | 6 |
| | Day 11 | 1.96 | 0.12 | 0.29 | 6 |
| 100 mg/kg cimetidine | Day 1 | 1 | 0.25 | 0.61 | 6 |
| | Day 2 | 1.05 | 0.3 | 0.73 | 6 |
| | Day 3 | 1.05 | 0.28 | 0.69 | 6 |
| | Day 4 | 1.25 | 0.3 | 0.73 | 6 |
| | Day 5 | 1.17 | 0.23 | 0.56 | 6 |
| | Day 6 | 1.37 | 0.26 | 0.64 | 6 |
| | Day 7 | 1.47 | 0.21 | 0.51 | 6 |
| | Day 8 | 1.73 | 0.16 | 0.39 | 6 |
| | Day 9 | 1.88 | 0.16 | 0.39 | 6 |
| | Day 10 | 2.4 | 0.15 | 0.37 | 6 |
| | Day 11 | 2.34 | 0.34 | 0.83 | 6 |

Stdev was calculated using formula SD = SEM*(SQRT n).

| Supplemental Figure 1: ACHN xenograft tumor size | | Normalized mean weight | SEM | SD | N |
|---|---|---|---|---|---|
| PBS | Day 1 | 1 | 0.09 | 0.22 | 6 |
| | Day 2 | 1.37 | 0.09 | 0.22 | 6 |
| | Day 3 | 1.39 | 0.09 | 0.22 | 6 |
| | Day 4 | 1.45 | 0.09 | 0.22 | 6 |
| | Day 5 | 1.39 | 0.09 | 0.22 | 6 |

*Continued on next page*

*Continued*

| | | | | | |
|---|---|---|---|---|---|
| | Day 6 | 1.52 | 0.08 | 0.20 | 6 |
| | Day 7 | 1.64 | 0.09 | 0.22 | 6 |
| | Day 8 | 1.84 | 0.09 | 0.22 | 6 |
| | Day 9 | 1.67 | 0.13 | 0.32 | 6 |
| | Day 10 | 1.92 | 0.08 | 0.20 | 6 |
| | Day 11 | 2.14 | 0.09 | 0.22 | 6 |
| 100 mg/kg cimetidine | Day 1 | 1 | 0.2 | 0.49 | 6 |
| | Day 2 | 1.26 | 0.14 | 0.34 | 6 |
| | Day 3 | 1.23 | 0.11 | 0.27 | 6 |
| | Day 4 | 1.1 | 0.1 | 0.24 | 6 |
| | Day 5 | 1.23 | 0.11 | 0.27 | 6 |
| | Day 6 | 1.34 | 0.09 | 0.22 | 6 |
| | Day 7 | 1.18 | 0.07 | 0.17 | 6 |
| | Day 8 | 1.34 | 0.09 | 0.22 | 6 |
| | Day 9 | 1.7 | 0.1 | 0.24 | 6 |
| | Day 10 | 1.7 | 0.08 | 0.20 | 6 |
| | Day 11 | 2 | 0.1 | 0.24 | 6 |

Stdev was calculated using formula SD = SEM*(SQRT n).

## Test family

- Two way ANOVA (2 × 2 factorial, PBS cohort and cimetidine cohorts only) followed by Bonferroni corrected Welch's *t*-tests for the following comparisons:
  1. PBS-treated A549 tumors vs cimetidine-treated A459 tumors.
  2. PBS-treated ACHN tumors vs cimetidine-treated ACHN tumors.
- Comparison of PBS-treated A459 tumors to doxorubicin-treated tumors.
  1. Bonferroni corrected Welch's *t*-test outside the framework of the ANOVA.

## Power calculations

- Power calculations were performed with R software 3.1.2 (R Core team, 2014) and G*Power (*Faul et al., 2007*).

**ANOVA; all groups at day 11 time point**

| F (1,20) (interaction) | $\eta_P^2$ | Effect size f | Power | Total sample size across all groups |
|---|---|---|---|---|
| 3.639500 | 0.153959 | 0.4265862 | 82.39% | 48 |

| Group 1 | Group 2 | Glass' delta* | $\alpha$ | A priori power | Sample size group 1 | Sample size group 2 |
|---|---|---|---|---|---|---|
| Bonferroni corrected Welch's *t*-tests | | | | | | |
| PBS-treated A549 at day 11 | Cimetidine-treated A549 at day 11 | 1.71429 | 0.0167 | 80.50% | 11† | 11† |
| Additional comparisons outside the ANOVA framework | | | | | | |
| PBS-treated A549 at day 11 | Doxorubicin-treated A549 at day 11 | 2.39286 | 0.0167 | 88.29% | 4† | 4† |

*The PBS control group SD was used as the divisor.
†With a sample size of 12 per group derived from the ANOVA, achieved power will be at least 84.36%.

## Acknowledgements

We thank Courtney Soderberg at the Center for Open Science for assistance with statistical analyses. We would also like to thank the following companies for generously donating reagents to the Reproducibility Project: Cancer Biology; American Tissue Type Collection (ATCC), BioLegend, Cell Signaling Technology, Charles River Laboratories, Corning Incorporated, DDC Medical, EMD Millipore, Harlan Laboratories, LI-COR Biosciences, Mirus Bio, Novus Biologicals, Sigma–Aldrich, and System Biosciences (SBI).

## Additional information

### Group author details

**Reproducibility Project: Cancer Biology**

Elizabeth Iorns: Science Exchange, Palo Alto, California; William Gunn: Mendeley, London, United Kingdom; Fraser Tan: Science Exchange, Palo Alto, California; Joelle Lomax: Science Exchange, Palo Alto, California; Nicole Perfito: Science Exchange, Palo Alto, California; Timothy Errington: Center for Open Science, Charlottesville, Virginia

### Competing interests

RP:CB: We disclose that EI, FT, JL, and NP are employed by and hold shares in Science Exchange Inc. The experiments presented in this manuscript will be conducted by IK at the Developmental Therapeutics Core, which is a Science Exchange lab. The other authors declare that no competing interests exist.

### Funding

| Funder | Author |
| --- | --- |
| Laura and John Arnold Foundation | Reproducibility Project: Cancer Biology |

The Reproducibility Project: Cancer Biology is funded by the Laura and John Arnold Foundation, provided to the Center for Open Science in collaboration with Science Exchange. The funder had no role in study design or the decision to submit the work for publication.

### Author contributions

IK, IZ, Drafting or revising the article; RP:CB, Conception and design, Drafting or revising the article

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
