## [Decision Letter]

Thank you for sending your work entitled “Registered report: Discovery and
preclinical validation of drug indications using compendia of public gene expression
data” for consideration at *eLife*. Your article has been
favorably evaluated by Stylianos Antonarakis (Senior editor), Chi Dang (Reviewing
editor), and 3 reviewers, one of whom is a biostatistician.

The Reviewing editor and the reviewers discussed their comments before we reached this
decision, and the Reviewing editor has assembled the following comments to help you
prepare a revised submission.

In this study, the authors propose a study to reproduce the findings reported in Figure
4C/D and Supplementary Figure 1 from a previously published manuscript (Sirota et al.
Sci Trans Med, 2010), which aimed at assessing the ability to predict drug repurposing
opportunities based on connectivity map data analysis. Specifically, the previous Sci
Trans Med paper reports that cimetidine, a histamine-2 (H2) receptor agonist commonly
used to treat peptic ulcers, can diminish lung cancer tumorigenesis in vivo. There are
several key concerns about the design of the study. The first concern is about the
duration of the experiment and statistical analysis, and the second about conclusions
drawn from using only one lung cancer cell line.

1) At the beginning of the Materials and methods section: The authors plan to follow the
mice for 11 days instead of 12 days. Is there a good reason to follow the mice one day
short? In addition, the experiment contains five cohorts. Among the five cohorts, cohort
2 only has 5 mice while the other 4 cohorts have 14 mice. Please justify.

2) Power calculation was based on *t*-test. It is suggested that the
authors use two-tailed unequal variance *t*-test if normality is not
violated or the use of Wilcoxon rank-sum test if normality is violated. The authors
propose the use of two-way ANOVA followed by *t*-test for analyzing tumor
weight data (in the subsection headed “Confirmatory analysis plan”).
Please make sure that the data do not violate the assumptions of ANOVA: normality and
homoscedasiticity. If the data do not fit the assumptions well enough, please try to
find a data transformation that makes them fit. If this doesn't work, please
apply a nonparametric counterpart of ANOVA such as Kruskal–Wallis test. In
addition, I suggest the use of contrast within the ANOVA framework instead of
*t*-test if the assumptions of ANOVA are met.

3) To compare growth curves of tumors, the authors propose ANCOVA followed by Bonferroni
corrected *t*-test. Please make sure that the data do not violate the
assumptions of ANCOVA and perform transformation or use non-parametric ANCOVA if
needed.

4) For the additional comparison of PBS-treated A459 tumors to Doxorubicin treated
tumors (in the subsection headed “Confirmatory analysis plan” and in the
subsection headed “Test family”), I suggest the use of two-tailed unequal
variance *t*-test instead of *t*-test if normality is not
violated or the use of Wilcoxon rank-sum test if normality is violated.

5) Although the reproducibility project is aimed toward reproducing previously published
results, the reviewers would like for the authors to address the limitation of drawing
conclusions for the use of only one cell line, A549. Specifically, activity of drugs in
cell lines and xenografts is generally highly idiosyncratic. As a result, most journals
require that any in vitro and in vivo experiments are replicated in multiple cell lines
and in vivo models.

---

## [Author Response]

*1) At the beginning of the Materials and methods section: The authors plan to
follow the mice for 11 days instead of 12 days. Is there a good reason to follow the
mice one day short? In addition, the experiment contains five cohorts. Among the five
cohorts, cohort 2 only has 5 mice while the other 4 cohorts have 14 mice. Please
justify*.

Figure 4C and Supplemental Figure 1 show data from Days 1 through 11. Although Day 12 is
displayed on the graphs, no data is present on that timepoint. We interpreted this to
mean that the original authors counted Day 0 as one of the days of monitoring; including
Day 0 accounts for the 12 days of monitoring mentioned by the authors and reconciles
that statement with the 11 days of data displayed in Figure 4D and Supplemental Figure
1.

For budgetary and ethical reasons, we wished to minimize the number of animals required
for these experiments. Thus, we did not use equal sample sizes for the additional
positive control (doxorubicin) treatment group when performing our power calculations.
These calculations demonstrated that we could use 3 mice in the control group and still
achieve 80% power to detect the original data’s effect size. The main aim of the
experiment is to test the effect of vehicle (PBS) treatment compared to experimental
(cimetidine) treatment, which includes an equal number of mice in each cohort.
Considering the unbalanced nature of this design we will be performing a planned
comparison between the PBS and doxorubicin outside the framework of the ANOVA, however,
the four cohorts that are the main aim of the experiment are analyzed within the
balanced ANOVA framework.

*2) Power calculation was based on* t*-test. It is suggested that
the authors use two-tailed unequal variance* t*-test if normality is
not violated or the use of Wilcoxon rank-sum test if normality is violated. The
authors propose the use of two-way ANOVA followed by* t*-test for
analyzing tumor weight data (in the subsection headed “Confirmatory analysis
plan”). Please make sure that the data do not violate the assumptions of
ANOVA: normality and homoscedasiticity. If the data do not fit the assumptions well
enough, please try to find a data transformation that makes them fit. If this
doesn't work, please apply a nonparametric counterpart of ANOVA such as
Kruskal–Wallis test. In addition, I suggest the use of contrast within the
ANOVA framework instead of* t*-test if the assumptions of ANOVA are
met*.

We have added language to the manuscript to clarify that we will perform the normality
and homoscedasticity tests. The original data was not shared, so instead summary
statistics estimated from the graph presented in Figure 4C and Supplemental Figure 1
were used. This limits what we can ascertain from the original data. We recalculated the
samples sizes using a Welch’s *t*-test instead of a
Student’s. The sample size we have planned is still sufficient. This is also true
for the non-parametric Wilcoxon rank-sum test. We plan to use the contrast within the
ANOVA framework if the assumptions are met, with the exception of the doxorubicin
cohort, which will be performed outside the framework because of the unbalanced
design.

*3) To compare growth curves of tumors, the authors propose ANCOVA followed by
Bonferroni corrected* t*-tests. Please make sure that the data do not
violate the assumptions of ANCOVA and perform transformation or use non-parametric
ANCOVA if needed*.

Pursuant to one of the later comments, we have removed the exploratory analysis by area
under the curve from the manuscript.

*4) For the additional comparison of PBS-treated A459 tumors to Doxorubicin
treated tumors (in the subsection headed “Confirmatory analysis plan”
and in the subsection headed “Test family”), I suggest the use of
two-tailed unequal variance* t*-test instead of*
t*-test if normality is not violated or the use of Wilcoxon rank-sum test if
normality is violated*.

We have added language to the manuscript to clarify that we will perform the normality
and homoscedasticity tests. As described earlier, we used summary statistics estimated
from the published data to perform power calculations. This limits what we can ascertain
from the original data. We recalculated the samples sizes using a Welch’s
*t*-test instead of a Student’s. The sample size we have
planned is still sufficient. This is also true for the non-parametric Wilcoxon rank-sum
test.

*5) Although the reproducibility project is aimed toward reproducing previously
published results, the reviewers would like for the authors to address the limitation
of drawing conclusions for the use of only one cell line, A549. Specifically,
activity of drugs in cell lines and xenografts is generally highly idiosyncratic. As
a result, most journals require that any in vitro and in vivo experiments are
replicated in multiple cell lines and in vivo models*.

Thank you for providing this insight. We have added a sentence addressing this point to
the Introduction. We will also include this point in the Discussion section of the
Replication Study that will be published once the replication data has been generated.
We agree the use of one model limits the overall conclusions that can be drawn. This
project focuses on understanding if the effects drawn from a single model can be
reproduced. While this does not speak to the robustness of the effect, such as can be
inferred through multiple models/approaches, it does provide a mechanism to examine the
extent to which an effect with a given model can be observed again. We will also limit
the conclusions that can be drawn to only this model.